# Why Re-Invent the Wheel? Social Network Approaches Can Be Used to Mitigate SARS-CoV-2 Related Disparities in Latinx Seasonal Farmworkers

**DOI:** 10.3390/ijerph182312709

**Published:** 2021-12-02

**Authors:** Mariano Kanamori, Daniel Castaneda, Kyle J. Self, Lucy Sanchez, Yesenia Rosas, Edda Rodriguez, Cho-Hee Shrader, Juan Arroyo-Flores, Ariana Johnson, John Skvoretz, Daniel Gomez, Mark Williams

**Affiliations:** 1Department of Public Health Science, School of Medicine, University of Miami Miller, Miami, FL 33136, USA; mkanamori@med.miami.edu (M.K.); d.castaneda@med.miami.edu (D.C.); Cilla39047@gmail.com (L.S.); yrosas@miami.edu (Y.R.); e.rodriguez6@miami.edu (E.R.); choshrader@gmail.com (C.-H.S.); juan.g.arroyo@gmail.com (J.A.-F.); alj74@miami.edu (A.J.); 2Department of Medicine, School of Medicine, University of Miami Miller, Miami, FL 33136, USA; d.gomez12@med.miami.edu; 3Department of Educational and Psychological Studies, University of Miami, Miami, FL 33136, USA; 4Department of Sociology, University of South Florida, Tampa, FL 33620, USA; jskvoretz@usf.edu; 5Fay W. Boozman College of Public Health, University of Arkansas for Medical Sciences, Little Rock, AR 72205, USA; MLW@uams.edu

**Keywords:** SARS-CoV-2, Latino, social networks, farmworkers

## Abstract

Latinx seasonal farmworkers are essential workers and are at elevated risk for SARS-CoV-2 in the United States. Risk factors for SARS-CoV-2 are unique to this population and include crowded living conditions, isolated social networks, and exploitative working environments. The circumstances and cultural values of Latinx seasonal farmworkers pose a unique challenge to public health authorities working to contain the spread of SARS-CoV-2. This community is in dire need of urgent public health research to identify opportunities to prevent SARS-CoV-2 transmission: social network methods could be the solution. Using previously collected and new information provided by a team of experts, this commentary provides a brief description of Latinx seasonal farmworker disparities that affect tracking and treating SARS-CoV-2 in this important group, the challenges introduced by SARS-CoV-2, and how social network approaches learned from other infectious disease prevention strategies can address these disparities.

## 1. Introduction

The SARS-CoV-2 pandemic has devastated entire communities, especially those already suffering from healthcare disparities due to structural racism in healthcare systems [1]. Health disparities have been exacerbated for marginalized populations during the SARS-CoV-2 pandemic, similar to in past pandemics in both the U.S. and internationally [2]. Among marginalized populations, seasonal farmworkers in the U.S. have been particularly devastated by SARS-CoV-2 and are at increased risk for being infected [3]. Specifically, seasonal farmworkers, composed of individuals, and family members, who are “employed in temporary farmwork but do not move from their permanent residence to seek farmwork”, often face health disparities related to their working status such as barriers to mental and physical health services (e.g., language, illiteracy), overcrowded working and living conditions that make transmission of the virus easier, and limited options for mental or physical healthcare in the event of getting sick and being unable to work [4]. In the United States, seasonal farmworkers are mainly male, Spanish-speaking, and Latinx. Additionally, this group suffers not only from physical health disparities, but also faces stressors that exacerbate mental health issues [5].

According to the Centers for Disease Control and Prevention (CDC), Latinxs are 4.6 times more likely to be hospitalized due to SARS-CoV-2 when compared to White Non-Latinxs [6]. SARS-CoV-2-related hospitalizations among Latinxs, and especially among Latinx seasonal farmworkers, may be especially high due to unique testing, vaccination, and treatment barriers. Previously identified healthcare accessibility factors are language and cultural barriers, health illiteracy, lack of insurance, lack of finances, undocumented immigration status, immigration stress, fear of deportation, and worry of healthcare-related discrimination [7,8,9]. SARS-CoV-2 may amplify these disparities. Due to these factors, and in accordance with the Latinx cultural value of *fatalism*, the Latinx seasonal farmworker community may view SARS-CoV-2-related treatment at healthcare centers as an extreme last resort, as hospitalization is either equivalent to death or deportation. When Latinx seasonal farmworkers do seek treatment, symptoms of SARS-CoV-2 may be severe enough to require hospitalization.

Using previously collected and new information provided by a team of experts (two members from the seasonal farmworker community, six Latinxs working in the seasonal farmworker community and other Latinx underserved communities, two mental health professionals, and six social network experts), we present a case example of how a marginalized group such as Latinx seasonal farmworkers have had increasing health disparities because of SARS-CoV-2, which may affect pandemic tracking, treatment, and promoting SARS-CoV-2 vaccines. The goal of this commentary is to outline challenges to pandemic tracking, treatment, and acceptance of vaccination among Latinx seasonal farmworkers and to explore how social network approaches have been successfully utilized among this community and may be tailored as a strategy to overcome the aforementioned challenges. The authors present this commentary for a global audience to consider how social network-based approaches may have utility in developing health programs for marginalized groups during periods of hardship such as the SARS-CoV-2 pandemic.

## 2. Challenges to Pandemic Tracking, Treatment, and Acceptance of SARS-CoV-2 Testing and Vaccination among Marginalized and Vulnerable Populations, Latinx Seasonal Farmworkers in Particular

### 2.1. Immigration Stress

Feelings of anxiety or fear of discrimination associated with healthcare settings may be affecting Latinx seasonal farmworkers’ willingness to be tested and vaccinated for SARS-CoV-2. Anecdotally, in part because of the “public charge” rule, Latinx seasonal farmworkers wanting to obtain American citizenship may think a positive SARS-CoV-2 test will negatively impact their immigration status or prolong the citizenship process [10,11]. Undocumented Latinx seasonal farmworkers may fear deportation if they go to a testing or vaccination center, and their undocumented status becomes known. Due to immigration stress, many Latinx seasonal farmworkers may avoid healthcare provider visits, which would enable them to access SARS-CoV-2 testing and vaccination.

### 2.2. Lack of Social Support

Prior to Hurricane Andrew, close extended families composed most of the Latinx seasonal farmworker community [12]. Since that time, many Latinx seasonal farmworkers no longer live around their extended families. Most mothers no longer raise children with the children’s fathers nor have other family members to help with childcare [12]. Since the closure of schools due to SARS-CoV-2, some children have been staying unsupervised at home, straining family finances to provide extra food and home-schooling supplies. Latinx parents may sometimes experience feelings of isolation due to the nature of seasonal work and immigration issues [13]. In addition to families, the church plays an important community role among Latinx seasonal farmworkers [13]. Connections made in church are important trusting relationships and for many seasonal farmworkers, the same people they work alongside in the fields and factories are the ones they sit beside at church, prayer meetings, or home bible study. The church is playing an important role in identifying and engaging families affected by SARS-CoV-2. This poses a unique exposure risk, considering how interconnected seasonal farmworkers are to one another. Despite changes in Latinx seasonal farmworker communities since Hurricane Andrew, families and friends are physically and emotionally tight knit, which makes practicing or enforcing social distancing a particular challenge. Some myths developing around SARS-CoV-2 in the community make treatment difficult. Due to the Latinx cultural value of *fatalism*, some families that have a member(s) diagnosed with SARS-CoV-2 may refuse to seek treatment for the fear that a hospitalization is equivalent to death.

### 2.3. Mistreatment at Work

Members from this community report “mistreatment” and “abuse” from work supervisors [13]. Some Latinx seasonal farmworkers have felt compelled to work even while showing SARS-CoV-2 symptoms because of fear of losing their jobs or because they cannot survive without the income. The fear of losing their job is especially strong when workers are supporting their families in the U.S. and their home countries. In the Latinx seasonal farmworker community, several families typically live in a single house and share bills to help pay the rent [13]. State and national financial-aid initiatives tend to be household-based, meaning that for families where household expenses are shared, applying for benefits means possibly having to share a small monetary check to offset needs between two or more families, if they qualify at all. Undocumented Latinx seasonal farmworkers do not qualify for unemployment benefits or food assistance, which makes taking time off work almost impossible. Many Latinx seasonal farmworkers do not own vehicles and are forced to carpool to work and elsewhere. Carpooling places workers in a vulnerable position by crowding them into small spaces for extended periods of time. Finally, some job sites in this community may not be following CDC-recommended safety protocols for their employees (e.g., hand sanitizer is not provided, social distancing is unenforced, and masks are optional and not provided).

### 2.4. Mental Health and Substance Use Disorders

The Latinx seasonal farmworker community is at increased risk for prescription drug and opioid misuse, greater alcohol use severity, and other substance use disorders. Participants in one of our previous studies reported prescription medication misuse as an escape from poverty, inhumane housing situations, and the physical and emotional abuse experienced in the workplace [13]. We found that legal and discrimination concerns are associated with increased at-risk drinking among women and emotional abuse with increase drug abuse among farmworkers. Drug abuse particularly might be associated with lower rates of SARS-CoV-2 testing and vaccination. Household members with drug abuse problems may not allow others in the household to be tested or vaccinated for SARS-CoV-2 because of fears of problems with law enforcement authorities. Those on home arrest for drug problems obviously cannot be tested for SARS-CoV-2 because they cannot leave their homes.

## 3. Social Network Approach as a Strategy to Overcome Challenges of Pandemic Tracking, Treatment, and Acceptance of SARS-CoV-2 Testing and Vaccination: A Case Example 

This commentary builds on lessons learned from our social network-based HIV prevention intervention called PROGRESO [14,15]. In this study, our team found that the vast majority of Latinx seasonal farmworkers live in fear due to the threat of deportation and discrimination [9]. Of the 260 participants we interviewed, half were worried about accessing immigration legal services, and two-thirds had been questioned about their status [9]. When interacting with the healthcare system, many participants reported feeling tense or discriminated against (46%) during an encounter with healthcare because of their inability to communicate in English (55%) or because of their culture (47%).

The lessons we have learned from PROGRESO could inform potential approaches for promoting SARS-CoV-2 testing, tracing, social isolation, treatment and vaccination [14,15]. Using community-based participatory research approaches, we identified a community partner located in a Latinx seasonal working community in South Florida to implement the study. It is important that recruitment and intervention procedures for SARS-CoV-2 testing and vaccination programs incorporate Latinx cultural values such as *personalismo* and *collectivism*. The cultural value of *personalismo* refers to a preference for friendship with close individuals with similar sociodemographic characteristics (i.e., homophily), suggesting a preference for familiarity in these relationships [16]. Thus, *personalismo* was incorporated into our social network recruitment in a way that reflected the preference for holding conversations about sensitive topics, such as HIV prevention, only after establishing friendships based on trust, support, and empathy [17,18]. Members from our community partner-identified twenty seed individuals, the “Latinx leaders.” Seeds were respected women in this community who had the ability to reach many peers. Recruitment procedures also incorporated the cultural value of *collectivism*. In Latinx communities, *collectivism* is a cultural orientation that values close, nurturing, and supportive interpersonal relationships over individualistic behaviors and attitudes. According to this orientation, a person is seen as part of a family and the broader community of friends and acquaintances and should accept responsibility for this role [19]. Consistent with *collectivism*, Latinx leader seeds can disseminate SARS-CoV-2 information within their social networks, so that underserved Latinxs in their community could support one another.

Current SARS-CoV-2 testing and vaccination programs could benefit from a link-tracing or social network respondent-driven sampling design. Our approach is based on groups of 13 participants, an adequate number for a social network-based intervention. Each seed was asked to invite three friends (i.e., first-order friends). The first-order friends were further asked to invite three of their own friends (i.e., second-order friends) [20]. As a result, for each seed, a network of 13 participants was recruited in three respondent-driven waves, with wave 1 including the seed, wave 2 including three first-order friends, and wave 3 including nine second-order friends. Seeds were asked to have all members of their social network contact the project coordinator. If any invited friend declined participation, the seed/friend was asked to invite a substitute friend until a social network of 13 participants was formed. We provided Latinx Leaders with additional instructions on how to promote interactions and conversations about HIV prevention within their friendship networks. The same approach can be used to promote SARS-CoV-2 information. These types of word-of-mouth recruitment strategies incorporating Latinx cultural values have been shown to work successfully in similar populations and settings [21].

The incorporation of *personalismo* in using a link-tracing design recruitment in our intervention allowed for the recruitment of groups of Latinx seasonal farmworkers who were friends or who had a friend in common [14,15]. This social network recruitment strategy promoted participants’ engagement [14,15]. Participants reported that they communicated among themselves via phone calls to remind each other about attending the intervention or scheduling an interview, and in some cases when a seed could not personally contact one network member, contact was made by their mutual friends, usually via social media [14,15]. Participants reported being interested in engaging in these interventions, not only for their personal benefit, but also to allow their friends to benefit from the intervention [14,15]. For instance, some participants gave their program incentives to other participants from their social networks who could not afford transportation to the intervention [14,15]. Our social network-based approach incorporated Latinx cultural values and created a social environment that allowed Latinx participants to feel comfortable discussing sensitive topics, collaborate among themselves, and develop social capital [14,15].

Social network approaches can also include network visualization in contact tracing activities and the promotion of SARS-CoV-2 testing and vaccination. The infectious disease gold standard of contact tracing has been updated by the Mayo Clinic and the CDC for SARS-CoV-2 outbreaks; however, these approaches rely on healthcare providers (HCPs) to conduct interviews with exposed patients and HCPs must decide whether they will need to reach out to household or known contacts or whether the client will independently notify their contacts [22,23]. To address this issue, the Florida Department of Health recruited public health students to support community contact tracing [24]. Because contact tracing is a time intensive endeavor for HCPs, social network mapping can assist HCPs with having a conversation about a patient’s social circle, who needs to be contacted to limit further spread. Social network mapping can also identify social network members who should be contacted to disseminate SARS-CoV-2 information [25]. These methods have been successfully employed in previous projects that sought to increase HIV prevention activities, such as condom use and HIV prevention knowledge [26].

In the context of the SARS-CoV-2 pandemic, social networks can act as a structural support for those who experience mental health issues [27,28]. Research in other areas points to resiliency of social networks as an important protective factor for adverse mental health issues [29]. Regarding mental health and social networks, frameworks have been developed in other marginalized communities to show the interrelatedness between health beliefs, health education, and cultural appropriateness of health behavior [30]. One such model is the PEN-3 model, which has been used to identify elements of the social network that help or hinder reception of health information. Social network mapping can expand on models such as this, to identify the elements of social networks that promote mental health changes, and thereby can be strengthened with interventions that focus efforts on those elements [30,31]. More specifically, it is not well understood how the relationships within social networks are characterized (e.g., directionality and strength of connection) and whether the characteristics of these relationships correlate with different issues in mental health (e.g., depression, anxiety), especially during periods where social isolation occurs.

Although our case study focuses on HIV prevention, social network methodology has been successfully utilized in other contexts [32,33]. Social network analysis should also be considered with its limitations in mind. Studies that have a goal to maximize network data quality (e.g., information about quality of relationships within ego-centric networks) may not be generalizable to other marginalized groups [34]. In our example, the concept of homophily was incorporated to build relationships between community partners and participants prior to having potentially uncomfortable conversations about HIV. However, homophily may also act as a confound in social network analyses when examining causal relationships within social networks [35].

During the SARS-CoV-2 pandemic, there is an opportunity to understand the role social networks have in the spread of healthcare information for testing, health beliefs, and vaccination. Looking to the future, mental and public health researchers must also expand from this current pandemic to other stressful situations that may occur with marginalized populations globally, such as climate change and its effects on migration [29]. By combining the integration of cultural values with social network approaches, we can understand how marginalized communities such as Latinx seasonal farmworkers function during a pandemic and investigate the potentially protective factors of community bonds that safeguard against difficult situations arising from events such as SARS-CoV-2.

## 4. Conclusions

This commentary documents SARS-CoV-2 healthcare disparities affecting the Latinx seasonal farmworkers community and discusses social network-based activities for future research and treatment approaches. Latinx seasonal farmworkers may avoid seeking SARS-COV-2 testing and treatment due to determinants that include their living conditions, transportation, mental health, and substance use disorders. Local efforts to provide protection from the SARS-CoV-2 epidemic to such vulnerable populations should be undertaken. Like HIV, SARS-CoV-2 is heavily stigmatized, resulting in a lack of discussion, the promotion of misinformation, and a lack of knowledge regarding healthcare options. In this commentary, we presented a social network approach that has been successful in engaging Latinx seasonal farmworkers for HIV prevention: why reinvent the wheel for SARS-CoV-2? Instead, we believe that social network approaches can highly engage underserved communities for SARS-CoV-2 interventions and understand the mental health implications that result from the stressors and situations caused by the pandemic. Once these maps of social networks are developed for a community, there is more research to be undertaken to analyze the directionality of the relationships within these networks and how they are affected during a pandemic or other large scale stressful events. With regards to SARS-CoV-2, members of the group may face stigma for trying to encourage others to get vaccinated or may not feel that they have the social status in a network to be taken seriously regarding health decisions or information; these types of questions can be answered by a combination of social network mapping and qualitative data to provide context on these social connections [36]. Social network mapping may explain how key persons influence the health of the social structure, and when pandemics affect these persons negatively, the resilience of the social networks in which they reside is also negatively impacted.

## Data Availability

Not applicable.

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
