# Peer review of "Why Re-Invent the Wheel? Social Network Approaches Can Be Used to Mitigate SARS-CoV-2 Related Disparities in Latinx Seasonal Farmworkers"

_ijerph, 2021, doi:10.3390/ijerph182312709_

Round 1

Reviewer 1 Report

This paper includes important information for researchers and service providers.  This paper is organized using  a research report format that may under emphasize the valuable commentary that the authors are making as well possibly not be noticed as a commentary. If indeed the authors want to present this as a commentary, the term commentary should be in the title of the paper.  This reviewer take away from the title and substantive write-up is that social network approaches should be central to the body of the paper, while the previous research conducted by the authors should be described as serving a context for the present paper.

I would recommend that in keeping with the title of the paper, Why Re-invent the Wheel? Social Network Approaches Can Be Used to Mitigate COVID-19-Related Disparities in Latino Seasonal Farm Workers, the authors re-frame the paper so that it conveys that it is a commentary and not a research report as is the case now with the headings used (Materials and Methods, Results, and Discussion) and reorganize and move the discussion section of the paper so that it is centrally presented and is the focus of the paper.  If the authors are willing, the following suggestions are made regarding the outline and organization of the paper:

I. Introduction

  • Cleary state that the purpose of the paper is that it is a commentary and that the focus of the commentary is to describe how social network-based approaches can be utilized to develop health programs to track, treat, and promote COVID-19 vaccines among marginalized and hard to reach populations.
  • Emphasize this strategy- social network-based approaches-can benefit researchers and service providers in addressing COVID-19 among Latino Seasonal Farmworkers and other marginalized and vulnerable populations because of the known challenges of reaching them for conducting research and providing services.   

II. Challenges to Pandemic Tracking, Treatment, and Acceptance of COVID-19 vaccines among marginalized/vulnerable populations, Latino Seasonal Farmworkers in particular

  • Instead of describing materials and methods and results of the previous study, after the Introduction section, this section would then list and describe these challenges (which are listed under the current results section: immigration stress, lack of social support, mistreatment at work, and mental health and substance use disorders.

III. Social Network Approach as a Strategy to Overcome Challenges of Pandemic Tracking, Treatment, and Acceptance of COVID-19 vaccines: A Case Example

  • This section would describe Social Network Theory and Approaches, any research supporting the use of social networks in prevention and intervention efforts.
  • Provide a case example: Here, the authors can describe their use of social network as a strategy in their research on HIV prevention and treatment, briefly describe the study and more importantly, describe how the authors used social network which they have described step by step in the discussion section.

IV. Conclusions and Recommendations

  • The final section would restate the purpose of the commentary as well as summarizing the authors’ recommendations.

In this re-organization, it is a matter of moving around content that is already available in this manuscript submitted for review.

Reviewer 2 Report

The manuscript is well-written and its main goal is appropriate given the current COVID-19 situation. The authors adequately share their experience on a previous project (PROGRESO) to provide their arguments. Despite the potential differences between a sexually-transmitted disease such as HIV and the COVID-19 pandemic situation, their social network approach is relevant and worth considering for future health interventions on such vulnerable groups.

My only comments to the authors are:

  1. Although the authors explain well their arguments and previous data focused on the social network approach, they do not seem to compare it against the current status (or even alternative approaches); this will enrich the discussion and provide even more solid arguments to support their suggestions.
  2. May be discuss a little more some potential limitations of applying principles tested on sexually-transmitted diseases on other public health situations (such as COVID-19).
  3. Minor comment: on line 248 the number "18" appears without a context; I believe it is a citation but please correct.
